# Cardiac Fibroblasts: Helping or Hurting

**DOI:** 10.3390/genes16040381

**Published:** 2025-03-27

**Authors:** Mohammad Shameem, Shelby L. Olson, Ezequiel Marron Fernandez de Velasco, Akhilesh Kumar, Bhairab N. Singh

**Affiliations:** 1Department of Rehabilitation Medicine, University of Minnesota, Minneapolis, MN 55455, USA; mshameem@umn.edu; 2Department of Biomedical Engineering, University of Minnesota, Minneapolis, MN 55455, USA; slolson24@wisc.edu; 3Department of Pharmacology, University of Minnesota, Minneapolis, MN 55455, USA; marro014@umn.edu; 4Department of Medicine, University of Minnesota, Minneapolis, MN 55455, USA; kumar786@umn.edu; 5Stem Cell Institute, University of Minnesota, Minneapolis, MN 55455, USA

**Keywords:** fibroblasts, cardiomyocytes, cardiac muscle, signaling, cardiac fibrosis, cardiomyopathy

## Abstract

Cardiac fibroblasts (CFs) are the essential cell type for heart morphogenesis and homeostasis. In addition to maintaining the structural integrity of the heart tissue, muscle fibroblasts are involved in complex signaling cascades that regulate cardiomyocyte proliferation, migration, and maturation. While CFs serve as the primary source of extracellular matrix proteins (ECM), tissue repair, and paracrine signaling, they are also responsible for adverse pathological changes associated with cardiovascular disease. Following activation, fibroblasts produce excessive ECM components that ultimately lead to fibrosis and cardiac dysfunction. Decades of research have led to a much deeper understanding of the role of CFs in cardiogenesis. Recent studies using the single-cell genomic approach have focused on advancing the role of CFs in cellular interactions, and the mechanistic implications involved during cardiovascular development and disease. Arguably, the unique role of fibroblasts in development, tissue repair, and disease progression categorizes them into the friend or foe category. This brief review summarizes the current understanding of cardiac fibroblast biology and discusses the key findings in the context of development and pathophysiological conditions.

## 1. Introduction

Fibroblasts are distinct cells with mesenchymal origin [1,2]. They mainly serve as signaling niche cells for tissue-resident cells, providing microenvironmental cues that regulate their behavior, including differentiation and self-renewal [3]. Seminal contributions from the work of Abercrombie and colleagues propelled the field of fibroblast biology forward by demonstrating the active role of these cells in wound healing, shifting the paradigm from passive tissue to active cellular involvement [4]. Using in vitro cell culture experiments, they showed that cardiac fibroblasts (CFs) have high migratory behavior by dynamically changing the adhesion complexes [5]. Their findings underscored the pivotal roles of CF in tissue repair and regeneration [4,5]. Subsequently, the discovery of myofibroblasts by Gabbiani and colleagues led to the identification of cells with intermediate features between fibroblasts and smooth muscle cells [6]. These studies illuminated the complex cellular dynamics underlying wound healing, particularly the involvement of contractile machinery in tissue remodeling [6]. Development in the field of the in vitro culture technique [7], as well as the 3T3 fibroblast cell line from mouse embryos, provided a readily available model system to study fibroblast biology in health and disease condition [8]. In the following sections, we review the key findings related to fibroblasts biology in the cardiac muscle.

## 2. Cardiac Fibroblasts: Origin and Function

### 2.1. Cardiac Fibroblasts Origin

During embryogenesis, the majority of fibroblasts are derived from the paraxial mesoderm and lateral plate mesoderm precursors, which ultimately give rise to fibroblast lineages [9,10]. In particular, the CFs are predominantly derived from the epicardium, the outermost layer of the heart [11,12]. Cells from the proepicardium detach and migrate to attach onto the beating ventricular surface, giving rise to the epicardium (Figure 1) [13]. These epicardial cells undergo an epithelial-to-mesenchymal transition (EMT) to give rise to epicardial-derived fibroblast progenitors (EDFPs), which then differentiate into CFs with subsequent integration into the developing myocardium [14]. Further, lineage tracing experiments showed that CFs are derived from both pro-epicardium and endothelium [15,16]. Other studies have shown that CFs residing in the interventricular septum and right ventricle originate from valve endothelial cells [17]. These cells utilize a related process called endothelial-to-mesenchymal transition (EndMT) to differentiate into endothelial-derived fibroblast progenitors (EDFPs) [18] (Figure 1). While previous studies have indicated that CFs constitute the majority of all the cells in the adult mammalian heart, recent studies using a genomics approach from adult human hearts suggest a more modest cellular composition of CFs ranging from 15 to 30% [19]. It has been shown that the CFs express unique markers like discoidin domain-containing receptor 2 (DDR2), platelet-derived growth factor receptor *α* (Pdgfr*α*), Tcf21, and vimentin [20,21,22,23], (Figure 1). Often, the expression of more than one of these markers is sufficient evidence to indicate the presence of fibroblasts [24]. Table 1 provides a detailed description of CFs-specific markers at various developmental stages. These studies provide valuable insights into the fate of CFs during cardiac development. Future research aimed at understanding the mechanism of interaction between other cardiac cell types including endothelial and hematopoietic lineages will further add their roles in these processes.

### 2.2. Cardiac Fibroblasts Function

The primary role of CFs is to maintain cardiac function and provide structural integrity as well as respond to various physiological and pathological stimuli [40]. During development, CFs secrete a variety of cytokine, chemokine, and growth factors to modulate the tissue microenvironment (Figure 1) [41]. At these stages, they secrete ECM proteins including fibronectin, laminin, collagen, and proteoglycan to help in organogenesis (Figure 1) [41]. CFs have a high self-renewable capacity that is critical for the cardiogenic program and for the remodeling of connective tissue [3,42]. They replicate and replenish their own population to maintain tissue homeostasis in the heart [43]. Loss of CFs results in the reduction in ECM deposition and hepatocyte growth factor (HGF) production which are required for the remodeling of connective tissue [42,44].

Developmentally, embryonic CFs express high levels of growth factors and ECM proteins such as periostin, fibronectin, and tenascin C compared to adult FB [45]. Further, co-culture experiments revealed that embryonic CFs induce the proliferation of nascent cardiomyocytes by secreting the fibronectin and type 3 collagen in paracrine fashion that supports cardiomyocyte growth and maturation [45]. Therefore, embryonic CFs create a microenvironment that supports cardiomyocyte growth and maturation (Figure 1) CFs can directly also interact with cardiomyocytes through gap-junctional proteins like connexins (e.g., Cx40, Cx43, and Cx45), facilitating intercellular communication [46]. Recent findings indicate that CFs play an essential role in the maturation of cardiomyocytes [47]. Another study led by Luis Hortells and co-workers showed that highly proliferative periostin (Postn)^+^ CFs are required for cardiac nerve development and cardiomyocyte maturation [48]. This was confirmed using transgenic GFP reporter mice showing that Postn^+^ lineage CFs at early postnatal days express proliferation genes while Tcf21+ CFs showed differential expression of ECM-related genes during a later time, suggesting the cardiac growth and maturation [48]. Single-cell RNA-seq (scRNAseq) data from murine hearts at different postnatal stages showed that switching the fibroblast subtype from neonatal to adult induces cardiomyocyte maturation [49]. Further, in vitro co-culture studies involving isolated neonatal CMs with adult fibroblasts showed significantly reduced CM proliferation with improved maturation benchmarks [49]. To understand how the CFs affect CM maturation, single-cell RNAseq (scRNAseq) studies of murine CFs from postnatal day 1 (P1) and P56 showed that secreted proteins from the CFs contributed significantly as a source to promote CM maturation [50]. These studies revealed a critical function for fibroblasts in CM proliferation and maturation in both murine and human models. While these findings support the role of CF in controlling the postnatal development of cardiac systems, additional studies are needed to decipher the crosstalk between CM and non-CM populations. These will facilitate the generation of more robust three-dimensional models for disease modeling and drug discovery [51].

Despite the importance of fibroblasts in the preservation of muscle tissue, improper activation of fibroblasts (myofibroblasts) leads to the overproduction of ECM components, causing a stiff ECM profile (Figure 2) [52]. In response to cardiac injury, fibroblasts rapidly adopt an activated phenotype (αSMA+) with increased proliferative potential and high deposition of a collagen-rich ECM. This irregular profile promotes scarring and fibrosis, leading to further consequences such as muscle dysfunction and disease progression (Figure 2) [52]. Studies have shown that the extent of fibrosis is dependent upon the collagen content and ECM deposition following cardiac injury response. It may vary in overall tissue architecture and can range from interstitial or compact to patchy or diffuse type, as discussed by de Jong et al. [53]. Studies have focused on targeting the activated fibroblasts to suppress the fibrotic response. For example, the ablation of activated fibroblast led to a cardio-protective effect [54]. Other studies have targeted signaling pathways to block the activation of CFs, such as a blockade of TGF-β signaling using small molecule inhibitors or an antibody-mediated blockade of BMP-signaling which led to the reduction in fibrosis and improved cardiac function [55]. It is proposed that initial activation of CFs occurs to prevent the injury-induced damage, however, over time, hyperactivation results in excessive ECM deposition and scarring [56]. Therefore, stimulation of these repair mechanisms is not always desirable in medicine. Many surgical repairs require implantations of foreign bodies, such as mechanical valve replacements, stents, and bone repairs [57]. In these situations, regulation and suppression of myofibroblast initiation is necessary to avoid complications associated with fibroblast activations [58]. Future research aimed at understanding the mechanisms involved in their stimulation to modulate fibroblast lineages precisely will aid not only better surgical outcomes but also provide insight into delaying disease progressions by regulating fibrosis [59].

## 3. Signaling in CF During Cardiac Pathogenesis

Crosstalk between CFs and cardiomyocytes through paracrine signaling controls the phenotype and behavior of both cell populations [60,61]. While multiple factors have been implicated in this intercellular crosstalk, the following section will focus on the major signaling pathways involved in the regulation of CF biology and adverse heart tissue remodeling following injury. These include platelet-derived growth factor receptors (PDGFRs), transforming growth factor β (TGF-β), wingless-related integration site (Wnt) signaling, and Hippo signaling (Figure 3).

### 3.1. Platelet-Derived Growth Factor Receptors (PDGFRs) Signaling in CFs

PDGFs are glycoproteins primarily involved in cellular migrations and proliferation, particularly for cells of a mesenchymal origin, such as fibroblasts (Figure 3A) [44,62]. They are secreted as a disulfide-linked homodimer of two A subunits (PDGF-AA), two B subunits (PDGF-BB), or heterodimers of subunits A and B (PDGF-AB). Tyrosine kinase receptors to these ligands comprise two monomers, PDGFRα and PDGFRβ, which form homodimers PDGFRαα (binds PDGF-AA, -AB, and -BB), PDGFRββ (binds PDGF-BB), or the heterodimer PDGFRαβ (binds PDGF-AB and -BB) (Figure 3A) [63]. PDGF signaling is a key regulator of fibroblast self-renewal through proliferation during developmental expansion, and its dysregulation can contribute to pathological conditions [44,64]. Loss of *Pdgfrα* in mice resulted in a deficit in cardiac fibroblast formation [31]. The binding of PDGF ligands with its receptors activates RAS/ MAPK and the AKT/PI3K pathway, which regulate the transcription of genes required for proliferation, survival, and differentiation (Figure 3A) [31,44]. Dysregulation of these pathways can contribute to diseases, including fibrosis [65]. Previous studies in mice showed that the overexpression of PDGF-C under the control of *α-MHC* promoter leads to progressive fibrosis and hypertrophy in male mice and lethal dilated cardiomyopathy in female mice [66]. Further, overexpression of PDFG-D in transgenic mice induces cardiac fibrosis and the proliferation of vascular smooth muscle cells [67]. It has been reported that PDGF induces cardiac fibrosis by activating the TGF-β1 upon introducing PDGF isoforms using adenovirus-mediated delivery [68]. Another study using transgenic mice showed that gain of function for PDGF-A and PDGF-B isoforms induces cardiac fibrosis [65] with up to an 8-fold increased cardiac size and lethality. Meanwhile, the induction of PDGF-B leads to focal fibrosis, less fibrosis, and moderate cardiac hypertrophy. Notably, blocking PDGFR-α/β using neutralizing antibodies in myocardial infarction mouse models shows decreased collagen deposition in the injured area [69]. Challenging this notion, a recent study indicates that PDGF-AB does not induce fibroblast proliferation but rather reduces myofibroblast differentiation and induces cells with a transcriptome distinct from myofibroblasts [70]. While interesting, these findings point to the need for in-depth analysis of these pathways in a context-dependent manner.

### 3.2. Transforming Growth Factor-β (TGF-β) Signaling in CFs

Previous literature showed that fibroblasts and myofibroblasts regulate the structure and function of cardiomyocytes either directly by cell contact or indirectly by ECM and the release of signaling mediators [71,72]. It is reported that TGF-β signaling plays an important role in the development of cardiac fibrosis and hypertrophy [60]. Further, several studies have indicated an elevated level of TGF-β in myocardial infarction and hypertrophic heart [73,74]. TGF-β cytokine is crucial for the development of CFs and their activation into myofibroblasts, which are key players in the fibrotic response of the heart to injury or stress (Figure 2) [74]. Upon activation of TGF-β receptors 1/2, along with Smad2/3, the translocation of Smad complexes into the nucleus is induce and it regulates the transcription of ECM genes (Figure 3B) [75,76,77,78,79]. Gene knock-out mice studies showed that the deletion of TGF-β receptors or Smad2/3 in CFs led to reduced fibrosis and hypertrophy in response to pathological stimuli [80,81]. Another study showed that Mkk4, a negative regulator of the TGF-β1 signaling, is associated with atrial remodeling and arrhythmogenesis. Selective inactivation of *Mkk4* in mice showed increased interstitial fibrosis, upregulation of TGF-β1 signaling, and dysregulation of matrix metalloproteases compared to control [82]. Notably, decreased levels of *Mkk4* were observed in human patients with atrial fibrillation, suggesting its association with increased production of pro-fibrotic molecules compared to a control [82]. Cartledge et al., showed that TGF-β participates in bi-directional signaling regulation between fibroblasts and cardiomyocytes and influences Ca^2+^ ion handling in the heart. They co-cultured cardiomyocytes with CFs (αSMA-negative) or myofibroblasts (αSMA-positive) and found that CFs increase Ca^2+^ transient amplitude while myofibroblasts reduce it through paracrine signaling. Using a TGF-β antagonist, they confirmed that these effects are due to dynamic signaling between the fibroblasts and cardiomyocytes [60]. Neutralizing TGF-β by an antibody reduces cardiac fibrosis by inhibiting CF activation [47]. A recent study showed that BAG3 knock-out CFs induce TGF-β signaling and a fibrotic response, confirming the TGF-β signaling mediated the contribution of BAG3 knock-out CFs to contractility and cardiac fibrosis [83]. These studies shed new light on the effects of TGFβ signaling on CF-CM interaction and their downstream processes, such as proliferation, maturation, and ECM regulation (Figure 3B).

### 3.3. Wingless-Related Integration Site (Wnt) Signaling in CFs

Initially, Wnt signaling was studied for its role in fetal development, but recently it has emerged as a versatile growth factor involved in maintaining tissue homeostasis and contributing to pathological conditions [84,85]. In the absence of a Wnt ligand, cytoplasmic β-catenin phosphorylated and was targeted for ubiquitination and proteasomal degradation (Figure 3C) [86]. In the presence of Wnt ligands, it binds to frizzled receptors and forms complexes with co-receptors (LRP5/6), which leads to the phosphorylation of LRP [87]. The activated Fz/LRP complex recruits disheveled (DVL), Axin, and GSK3, leading to the sequestration of Axin and GSK3 at the plasma membrane. This allows β-catenin to translocate into the nucleus and induce transcription of ECM genes (Figure 3C) [88,89,90,91,92]. Previously, it has been shown that Wnt/β-catenin signaling synergizes with TGF-β signaling to induce the differentiation of fibroblasts into myofibroblasts, which are characterized by an increased expression of α-smooth muscle actin (α-SMA) and enhanced ECM production [93]. Xiang et al. showed that the deletion of β-catenin in CFs reduces interstitial fibrosis by reducing ECM deposition, and that the reduction in cardiomyocyte hypertrophy following pressure overload in mice [94]. Disruption of Wnt/βcatenin signaling in CFs decreases cardiac performance and affects wound healing after cardiac injury [95]. Further, increasing the canonical Wnt1 signaling cascade with the increase in HO-1 and adiponectin improves fibrosis and cardiac dysfunction in a mouse model of myocardial infarction. [96]. Many studies showed that canonical WNT/β-catenin and Smad-dependent TGF-β signaling induce myofibroblast activation and promote fibrosis [97,98]. It is reported that Wnt3a promotes the expression of αSMA in the cultured fibroblasts, which is reversed by the knock-down of β-catenin, suggesting these changes were dependent on canonical Wnt signaling through β-catenin [99]. Further, GSK-3β KO mice showed that the deletion of fibroblast-specific GSK-3β induces fibrosis by activating canonical TGF-β1-SMAD-3 signaling [100]. A recent study showed that TEAD1 promotes the fibroblast-to-myofibroblast transition via the Wnt signaling pathway. Genetic knock-down of Wnt4 inhibits the transformation of CFs and confirms the role of Wnt signaling in cardiac fibrosis [101]. Identifying additional mechanisms for controlling cardiac fibrosis will provide new strategies for mitigating cardiac damage and promoting repair.

### 3.4. Hippo Signaling in CFs

The Hippo pathway plays a crucial role in the proliferation and differentiation during cardiac development through Wnt or PI3K-AKT signaling pathways [102,103,104]. Hippo signaling cascade controls the activity of transcriptional coactivators YAP and TAZ (Figure 3D) [105]. Phosphorylation of YAP and TAZ induces polyubiquitination and subsequent degradation by the proteasome [106]. When the Hippo pathway is inactive, YAP and TAZ are dephosphorylated and translocate to the nucleus. They interact with transcription factors and other DNA-binding proteins to regulate the transcription of target genes (Figure 3D) [55,107]. Dysregulation of the Hippo pathway leads to cardiac diseases like cardiomyopathy, heart failure, coronary heart diseases, and myocardial infarction [108,109]. A previous study identified that YAP and TAZ proteins were significantly upregulated in fibrotic ECM tissue compared to healthy tissue [110]. Another study in mice showed that deletion of the Hippo pathway component Salv induces a reparative genetic program with increased vascularity in the scar regions, reduced fibrosis, and recovery of pumping function compared with controls [111]. Importantly, deletion of fibroblast-specific YAP/TAZ in mice reduces fibrotic and inflammatory response and improves cardiac function after myocardial infarction [112].

Other pathways include the IL-6 trans-signaling-STAT3 pathway, which mediates hypertrophic ECM and cellular proliferation [113]. In addition, fibroblasts utilize both angiotensin 2 (Ang 2) and cytokine signaling to initiate ECM contraction. For example, modified cytokine signaling affects specific cadherin proteins, which facilitate fibroblast–fibroblast connections and potassium channels, in turn deregulating Ang 2 and thus creating a feedback loop [114]. Finally, fibroblasts use other signals, such as myogenic progenitor cells (MPCs) and exosomes. For example, MPCs secrete certain exosomes that regulate fibroblast collagen expression by repressing Rrbp1 [115]. While there is still much to learn about the cascading pathways connecting fibroblasts, the existing knowledge provides a substantial framework regarding fibroblast biology and its regulation via the signaling pathways in development and pathophysiological conditions. The next section of this review will center around the role of CFs in cardiac diseases, including hypertrophic cardiomyopathy (HCM), dilated cardiomyopathy, and Duchenne muscular dystrophy (DMD).

## 4. Cardiac Fibroblasts in Cardiovascular Diseases

CFs are activated in response to pathological injuries such as myocardial infarction or in response to paracrine factors stimuli [116,117]. This leads to a phenotypic shift where CFs acquire characteristics of both fibroblasts and smooth muscle cells, including the expression of α smooth muscle actin (αSMA) [43]. Dysregulation of CF function results in abnormal myocardial architecture with alterations in the mechanical properties of the heart, which ultimately leads to disruption of mechano–electric coupling between cardiomyocytes [118]. Activated fibroblasts aid in disease progression in diseases such as DMD, HCM, and DCM [119,120,121]. Table 2 provides a summary of phenotypic changes in the CFs populations in these disease conditions.

### 4.1. Cardiac Fibroblasts in Hypertrophic Cardiomyopathy

Hypertrophic cardiomyopathy (HCM) is one of the most common congenital heart diseases in the world, with a prevalence rate of roughly 1 in every 500 individuals [122]. Additional late-stage phenotypic characteristics include myocyte hypertrophy, interstitial fibrosis, and overall cardiac dysfunction [125]. Prolonged symptoms result in more severe consequences, including atrial fibrillation, stroke, heart failure, blood clots due to compensatory chamber enlargement, and eventually sudden cardiac death [122].

It is widely known that mutations in the sarcomeric genes including, myosin heavy chain 7 (*MYH7*), myosin binding protein C (*MYBPC3*), cardiac troponin T (*TNNT2*), and cardiac troponin I (*TNNI3*) genes result in HCM disease [134,135]. A primary characteristic of HCM is myocyte disarray, hypertrophy, and ECM compositional changes, including accumulated fibrosis throughout the cardiac tissue. Studies have focused on various factors that could affect cardiac fibroblasts, thus increasing fibrosis during the HCM disease process [120]. More recent studies have focused on the impacts of TGF-β signaling on CFs secretion to determine the key factors in promoting fibrosis [83,136]. Studies have shown that CFs have activated TGF-β signals that initiate hypertrophy via the modification of the ECM through the enhanced secretion of IGF1 [137]. They proved this by controlling TGF-β signaling by administering a specific blocking antibody and found decreased IGF1 expression and a reduced fibrotic response [137]. Additional research into this signaling pathway confirmed its role in HCM development. Zou and co-workers utilized a CRISPR-mediated disruption technique to manipulate the *MYBPC3* gene and measured its effects [138]. They found the upregulation of TGF-β increased the hypoxia-inducible factor-1 subunit expression and *GLUT1*, *PFK*, and *LDHA*, which led to increased ATP biosynthesis production, which activated CFs promoting fibrosis [138].

As mentioned previously, the ECM is altered as HCM progresses. Therefore, pinpointing these changes is essential in understanding the early mechanistic changes in this disease. For example, impaired sarcomere function is correlated to the reduction in ECM cognate ligand secretion and dysfunctional communication between cardiac fibroblasts, lymphocytes, and integrin β1 [139,140,141]. These factors alter ECM composition, thus promoting HCM disease progression [141]. Another study dove further into the effects of integrin β1 and cognate ligand expression and discovered a plethora of cellular modifications [142]. They detected increases in cell communications, such as neuron to leukocyte, fibroblast to leukocyte, and dendritic cells, as well as overall endothelial cell communication through these altered expressions [141,143]. Dysfunctional communication affects the ECM and the binding of growth factors, integrins, PDGF, SMAD, and adenylate cyclase [68,141]. In addition, it caused calcium channel inhibitor activity and serine-threonine kinase activity in non-obstructive HCM [144]. A recent study showed that the MYH7 mutant HCM variant induced collagen deposition and tissue stiffness by altering the expression of paracrine factors, ECM, and inflammatory signaling [145]. Thus, it is apparent that CFs lead to the detrimental effects of HCM. Additional studies are needed to identify the altered pathways responsible for these changes and determine effective therapies to delay progression or prevent HCM disease.

### 4.2. Cardiac Fibroblasts in Dilated Cardiomyopathy

The cause of dilated cardiomyopathy (DCM) is much broader than hypertrophic cardiomyopathy [146]. In addition to genetic mutations in the sarcomere, DCM can evolve from mutations in genes that encode nuclear envelope and cytoskeletal proteins [147,148,149]. Additionally, DCM can appear due to causes unattributed to genetic modifications, including myocarditis, prolonged exposure to alcohol, drugs, and toxins, as well as disturbances affecting the endocrine system and metabolic pathways such as thyroid disease, diabetes, and even some pregnancies [150]. Depending on the source and severity of DCM development, some forms are reversible, a characteristic not available in HCM [151]. DCM is characterized by contractile dysfunction due to ventricular dilatation instead of the stiffening and thickening of the left ventricular wall, as in HCM [152]. This dilatation typically originates in the left ventricle and eventually causes remodeling in the right and atria. The enlargement of these chambers reduces contractile function, eventually leading to fluid build-up in the lungs and body, otherwise known as heart failure [153]. These pathological changes cause fatigue, leg swelling, fainting, weakness, cough, shortness of breath, and arrhythmias [153].

There are multiple steps to treat DCM, depending on the underlying cause [154]. However, if genetic in origin, the damage cannot be reversed and only be managed using diuretics, ACE inhibitors, and β-blockers [155,156,157]. Many consequences of DCM are similar to HCM, including stroke and sudden death [158,159]. Additionally, specific pathological changes and evidence of scar tissue or fibrosis are identical due to the ventricular remodeling [152]. Thus, much research has been conducted on fibroblasts’ role in the progression of dilated cardiomyopathy.

Current research identified an essential regulator of hypertrophic gene expression in cardiac muscle, the bromodomain and extra terminal family (BET) [160]. A recent study determined that BRD4, a transcription factor that is a part of the BET, was able to regulate proinflammatory gene expression in CFs [161]. Therefore, this suggests that BET inhibition could reverse specific inflammatory effects and fibrosis in DCM [162,163]. Previous research found a mitochondrial protein involved in both the development of hypertrophy and collagen secretion of neonatal cardiomyocytes [164]. Mutation of the protein tafazzin was later discovered to cause DCM in conjunction with another disease known as Barth syndrome [165]. However, there has not been much research on the effects of this protein in DCM.

Laminopathy or the mutation of *LMNA* (lamin A/C) also results in DCM phenotype [148]. Under hypoxic conditions, these cardiomyocytes showed altered calcium dynamics and similar pathological changes to DCM, thus making it worthwhile to analyze the cellular mechanisms responsible for this disease [166]. With this in mind, fibroblasts of mice with this condition showed uniquely methylated regions that improved distal features and repressed transcriptional chromatin [167]. More recently, LMNA-modeled DCM revealed that expression of a structurally altered connexin generates dysfunction in gap-junction communication through several modified mechanisms [168]. These mechanisms decreased the expression of *Hf1b/Sp4*, upregulated TGF-β signaling, and increased apoptosis due to an enlarged DNA damage response [169].

Another method of instigating a DCM phenotype is by infecting mice with the cardio-virulent coxsackie virus B3 which was associated with modifications to the ECM [170,171]. They found that this virus-induced DCM phenotype led to the upregulation of *elastin*, *collagen 1*, *collagen 3*, *chondroitin sulfate proteoglycan*, *plasminogen activator*, and *matrix metallopeptidases*, while simultaneously downregulating the tissue inhibitor of metalloproteinases [172]. These components confirmed DCM involvement in altering ECM pathways, impairing the ability to stabilize cellular activity, synthesize through fibroblast, regulate collagen deposition, and maximize cell adhesion [172].

Previous studies showed that increased *XT-I* and *proteoglycan* expression stimulate myocardial remodeling in DCM [173]. They found that TTβ signaling in concurrence with stress-induced *XT-I* expression promoted extracellular matrix remodeling in the dilated heart [173]. Further study into the effects of TGF-β on CFs revealed a nitro-oleic acid with anti-fibrotic effects in mice [174]. Utilizing the inhibition of phosphorylation to TGF-β downstream targets, they found that nitro-oleic acid can prevent myofibroblast trans-differentiation [175]. Another pathway involved in the initiation of DCM is the YAP signaling pathway [111,176]. It is a regulator of cardiac myofibroblast differentiation and thus has been the center of current research. Studies surrounding this pathway determined that activated YAP signaling could decrease differentiation, thus inhibiting ECM remodeling [177].

Studies like the ones mentioned above have initiated further research into the effects of extracellular matrix remodeling. Recently, atomic force microscopy displayed unaltered characteristics of CFs isolated from DCM hearts, including cellular viscosity, stiffness, and fluidity [178]. Yet, when healthy cardiomyocytes were co-cultured with CFs isolated from DCM hearts, these cardiomyocytes exhibited impaired diastolic functions and reduced contractility due to humoral factors and direct cell–cell contact [178]. These factors and the signaling pathways involved were further investigated using pathway analysis. They found increased activity of ECM–receptor interactions, focal adhesion signaling, Hippo signaling, and TGF-β pathway [178]. They also evaluated the mRNA expression of *collagen*, *fibronectin*, *α-smooth muscle actin*, and alternate TGF-β related genes in CFs isolated from DCM hearts [178]. They found altered expression compared to the control group [178]. Thus, it is clear that CFs and their role in altering ECM composition play an essential role in DCM disease progression. Future research is necessary to determine the exact components, signaling pathways, and therapies that will efficiently reverse or prevent dilated cardiomyopathy disease progression.

### 4.3. Cardiac Fibroblasts in Muscular Dystrophy

Muscular dystrophies are a category of muscle-wasting diseases caused by genetic mutations [179]. The severity of these diseases varies widely depending on the locus of the mutation [179]. They can range from minimal clinical symptoms to immobility and premature death. One category of muscular dystrophies is known as dystrophinopathies, which results from mutations in the *Dystrophin (DMD)* gene, the largest gene in the human genome, which encodes the protein dystrophin [129]. Dystrophin is essential in maintaining striated muscle structural integrity and acts as a shock absorber [180]. Without a functional dystrophin, muscles are damaged every time they contract. With advancing age, patients experience a decrease in the resilience of their muscles, impeding their ability to repair after injuries [180].

The four known dystrophinopathies are Duchenne muscular dystrophy (DMD), Becker muscular dystrophy (BMD), an intermediate pathology between DMD and BMD, and DMD-associated dilated cardiomyopathy [181]. DMD and BMD are the two most common pathologies and affect both striated muscle tissues [182]. They are caused by deletions of one or several exons in the *DMD* gene [129]. Less severe, in-frame mutations typically create truncated, or shortened, dystrophin, resulting in a milder form of this disease, BMD [183]. DMD occurs when the deletions are more severe and cause devastating frameshift mutations, sometimes eliminating dystrophin [130]. DMD is a rare disorder with a prevalence rate of roughly 1 in every 3500 males [184]. Although some females inherit the disorder, it primarily affects males due to its recessive nature and its location on the X chromosome [184]. Therefore, females are statistically more likely to be disease carriers than physically affected [185].

DMD is often diagnosed in early childhood, around 2–3 years of age, due to noticeable developmental delays [186]. The disease affects the proximal muscles first, causing weakness and muscle wasting in the child’s upper arms, shoulders, thighs, and hips [129,181,186]. Following the observation of these symptoms, the disease can be diagnosed using specific medical tests such as blood tests, muscle biopsies, electromyograms, and electrocardiograms [187,188,189]. Electromyograms are necessary to test the origin of muscle weakness and determine if the cause is a genetic mutation or nerve damage, while electrocardiograms monitor abnormal heart rhythms [190]. However, these arrhythmias are often seen much later in disease progression as the heart becomes affected. The muscle-wasting continues with age, affecting more imperative muscles such as the diaphragm and the heart [136]. Respiratory distress occurs as the diaphragm is affected, causing a significant decline in quality of life [191]. Eventually, cardiac distress overtakes the body as other symptoms, such as cardiomyopathy, begin to develop, commonly leading to premature death [185].

As mentioned above, DMD progresses through constant muscle tearing resulting in pseudohypertrophy, or unusual enlargement of specific muscles [192]. Additionally, fibrosis occurs with the accumulation of extra ECM components [193]. Considering this, a significant amount of research regarding DMD progression is focused on the influence and modification of fibroblasts [194]. Previous research discovered that DMD fibroblasts significantly differ in their traits and cascading functions [195]. For example, a study in 2011 found that fibroblasts affected by DMD can increase adhesive properties, are less vulnerable to cell death, and have a higher propensity for migration [194]. Additionally, modifications in the expression of FAK and ERK/MAPK were observed [194]. Recent developments showed that the exosomes secreted by DMD muscular fibroblasts phenotypically convert fibroblasts to myofibroblasts, causing a more excellent fibrotic response [196]. This study confirmed that this response resulted from an increase in *miR-199a-5p* and a decrease in caveolin-1 [196]. Further research indicated that CFs in the DMD heart had impaired actin microfilaments, altered metabolic pathways, and enhanced glycolysis [195]. In addition, these fibroblasts exhibited spatial mitochondrial rearrangements and a reduced mitochondrial number and respiratory function. Furthermore, these DMD hiPSC CFs had characteristics that resembled a myofibroblast phenotype and an increased susceptibility to pro-fibrotic stress [195]. Overall, while significant progress has been made in determining the role of fibroblasts and fibrosis in DMD, effective therapeutic techniques to slow disease progression are lacking.

## 5. Strategies to Mitigate Cardiac Fibrosis and Therapeutics

In the previous sections, we provided a detailed understanding of the physiological or pathological mechanisms of cardiac tissue remodeling during development and disease progression. In this section we describe the strategies to mitigate cardiac fibrosis.

### 5.1. Pharmalogical Interventions for Cardiac Diseases

The current therapies mainly target the pro-fibrotic agents involved in the differentiation of fibroblasts into myofibroblasts to attenuate the development of cardiac fibrosis in damaged hearts [197]. These therapies include but are not limited to, drugs focused on the Renin–Angiotensin–Aldosterone system (RAAS), ACE inhibitors, β-blockers, endothelin antagonists, statins, growth factors, and cytokines [198]. A recent study showed that sacubitril/valsartan, a novel angiotensin receptor inhibitor, can reduce cardiac remodeling by blocking angiotensin II type 1 receptors and increase vasoactive peptides through neprilysin inhibition [199]. Another drug, Eplerenone, which was approved by the FDA in 2002, was developed to inhibit fibrosis formation by targeting and blocking the aldosterone pathway [200]. Other therapies targeting cardiac fibrosis include strategies to inhibit fibroblast activation by blocking TGF-β or Smad3 and other signaling pathways [201,202]. Promising approaches involve the use of inhibitors such as CTGF mAb, pirfenidone, tissue nonspecific alkaline phosphatase (TNAP) inhibitor, baicalin, and CTRP9, which target various signaling pathways implicated in the progression of fibrosis in cardiac tissue [201,203]. In addition to TGF-β and CTGF inhibition, other strategies such as targeting collagen synthesis are also being investigated [204]. Several studies have explored other factors such as the glycoprotein endoglin in controlling myocardial fibrosis [205]. Endoglin also targets pathways that drive EndMT to prevent fibroblast accumulation from endothelial cells [206].

In the past decade, biomaterial-based strategies have emerged as promising approaches for addressing cardiac fibrosis [207]. Injectable hydrogels, which mimic the ECM, provide structural support while delivering anti-fibrotic drugs or growth factors to reduce fibrosis and encourage cardiac repair [207,208,209]. Additionally, encapsulating functional molecules like *mRNA-29B*, siRNA, miRNA, growth factors, and small molecules in scaffolds has demonstrated the potential to restore collagen balance in cardiac tissue [210,211,212]. Multifunctional biomaterials that combine drug delivery, structural support, and ECM-mimicking properties represent a comprehensive approach to treating CFs and restoring cardiac function [213].

### 5.2. Cell–Gene-Based Therapeutic Approach for Cardiac Diseases

Expanding on current strategies, the emphasis has shifted to recent advances in gene-based technologies, particularly mRNA and CRISPR-based approaches, which have opened new avenues for targeting cardiac fibrosis (Figure 4). In the recent past, CFs have been directly reprogrammed using cardiogenic factors including Gata4, Hand2, Myocardin, and Tbx5 (GHMT) to repair the fibrotic lesions following cardiac damage [214] (Figure 4). The methods for direct reprogramming of CFs into cardiomyocytes are elegantly reviewed elsewhere [215]. Similarly, other strategies have been used for targeting activated fibroblast. For example, several microRNAs (small non-coding RNA molecules) are classified as pro- or anti-fibrotic according to how they affect fibrosis [216,217]. Pro-fibrotic microRNAs, such as *miR-27b*, *miR-96*, and *miR-99b-3p*, promote fibroblast activation and ECM accumulation, contributing to myocardial fibrosis development [218,219]. On the other hand, anti-fibrotic miRNAs, like *miR-489, miR-1954,* and *miR-150*, may have therapeutic value by blocking fibrosis-related pathways and indicators [220,221,222]. Targeting these microRNAs to restore equilibrium between fibrosis-promoting and fibrosis-inhibiting activities is one tactic for lessening the severity of cardiac fibrosis. Similarly, CRISPR/Cas9 technology has shown promise as a potential treatment for cardiac fibrosis, with several studies demonstrating its effectiveness in preclinical models [223,224,225]. For example, CRISPR/Cas9 systems have been used to deactivate pro-fibrotic microRNA, *miR-34a*, to reduce fibrosis and improve heart function [224,225]. Furthermore, these tools have been utilized to reprogram fibroblasts into cardiac progenitor cells to promote tissue regeneration and reduce scar formation [226]. Similarly, CRISPR/Cas9-mediated overexpression of *IL-10* or integration of the *LEF1* gene in the stem cells has been shown to improve their survival with significantly reduced fibrosis and regenerative benefits, hence, providing a new approach for advanced cardiac therapies [227,228,229].

Additionally, recent advances in immunotherapy have emerged as a potent strategy to ablate myofibroblast [239]. Besides tumor immunotherapy, chimeric antigen receptor T (CAR-T)cell therapy has been investigated for its potential for suppressing cardiac fibrosis (Figure 4) [239]. Targeting molecules that are exclusively expressed in the pathological cells with minimal or no expression in healthy tissues is key for successful immunotherapy. Recent studies have identified fibroblast activation protein (FAP), a type II transmembrane serine protease, as a promising target [240]. *FAP* is expressed selectively in the CFs of the diseased hearts, such as HCM and DCM, and is rarely found in other cell types or healthy adult heart tissues [241]. Injection of engineered FAP-CAR-T cells successfully reduced fibrosis in fibrotic mouse hearts following injury [231]. Subsequent studies have used lipid nanoparticle (LNP)-encapsulated mRNA to generate FAP-CAR-T cells in vivo. In this study, researchers successfully delivered FAP-CAR mRNA to T cells in mice by combining it with CD5-modified LNPs [55]. LNPs facilitated mRNA internalization into T cells, allowing for direct synthesis of FAP-CAR-T cells to target myofibroblast and reduce cardiac fibrosis [233]. These findings show that CAR-T cell therapy targeting myofibroblasts has enormous potential for treating cardiac fibrosis.

Additionally, other alternative immunotherapies such as CAR-macrophages (CAR-M) and natural killer (CAR-NK) cells are being developed. Unlike CAR-T cells, CAR-M and CAR-NK cells have better infiltration abilities, allowing them to effectively navigate dense tissues [242]. Gao et al. recently found that transferring bone marrow-derived mononuclear with the CAR strategy for phagocytosis (CAR-P) to target FAP+ myofibroblasts effectively reduced cardiac fibrosis and improved heart function following heart injury in mice [215]. These findings provide proof-of-concept for using CAR-M cells to treat cardiac fibrosis. With ongoing technological advancements and extensive empirical research, CAR-NK/M therapies show great promise as effective treatments for cardiac fibrosis, giving patients new hope.

## 6. Concluding Remarks

CFs are specialized cell types that are essential for cardiogenesis and tissue homeostasis. During development, they mainly have autocrine and paracrine functions to regulate the overall tissue architecture. Following cardiac damage, CFs are hyperactivated leading to excessive ECM deposition and fibrosis. Targeting the hyperactivation of the cardiac fibroblast is the key to designing novel therapies for cardiac tissue damage. Since CFs are needed for all aspects of organ development, and their initial attempt is to repair the damaged tissue, we propose categorizing them as friends rather than foes.

## Figures and Tables

**Figure 1 genes-16-00381-f001:**
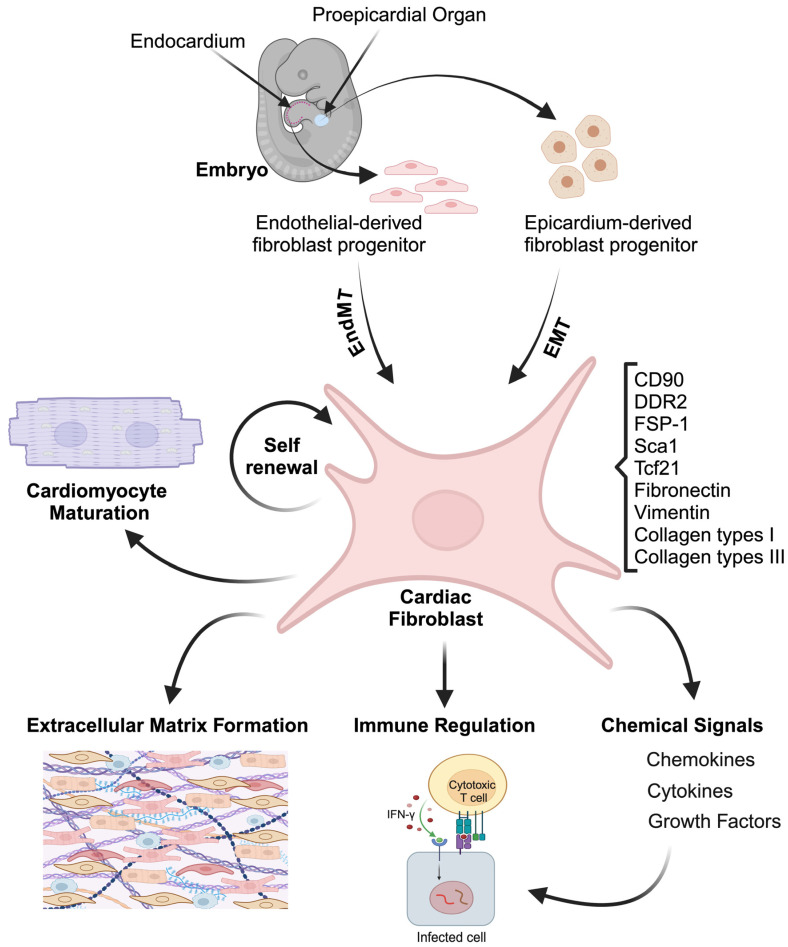
**Cardiac fibroblast origin and function.** CFs are crucial structural cells for the proper development of heart tissue. During embryonic development, the proepicardium organ gives rise to the epicardium-derived fibroblast progenitor cells through a process called epithelial–mesenchymal transition (EMT), and through endothelial by an endothelial–mesenchymal transformation (EndMT) process they are transitioned into cardiac fibroblast. CFs are characterized by the expression of unique markers such as CD90, DDR2, Sca1, FSP1, Fibronectin, Vimentin, Collagen Type I, and Collagen Type III. Cardiac fibroblasts play several key roles in maintaining tissue structure, function, and repair. One of the major roles of CF is to provide structural support to the developing organ. CFs synthesize the ECM by depositing collagen fiber, proteoglycans, elastin, fibronectin proteins, and laminins, and remodel the ECM by covalent crosslinking, protein glycosylation, as well as by secretion of matrix metalloproteinases (MMPs). Interactions between CM-CFs have led to increased cardiomyocyte maturation. Further, CFs play an important role in paracrine signaling by secreting a number of chemical signals including cytokines (TNFα, IFNγ, IL-6), chemokines (MMPs), and growth factors. Multiple studies have shown their role in immune regulation and inflammation following injury.

**Figure 2 genes-16-00381-f002:**
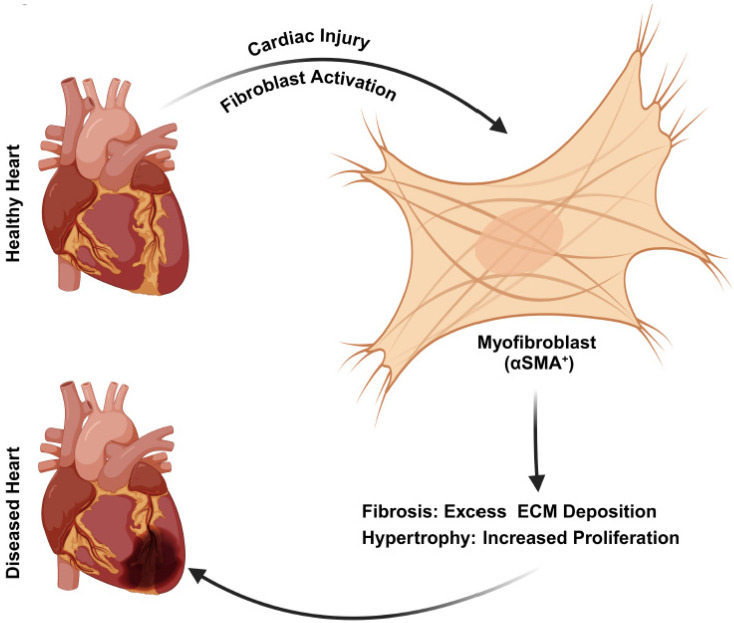
**Cardiac fibroblast activation in response to injury.** Resident fibroblasts are integral components of healthy tissues, actively participating in maintaining tissue structure, function, and homeostasis. Upon mechanical stress or tissue injuries, TGF-β and AngII signaling pathways are activated, activating resting fibroblast into myofibroblast. Myofibroblasts are specialized cells with high αSMA expression. These cells increased the production of ECM proteins such as collagen, fibronectin, and other microfibrillar proteins. Myofibroblasts play a crucial role in tissue repair but can also induce pathological fibrosis when their activation is dysregulated.

**Figure 3 genes-16-00381-f003:**
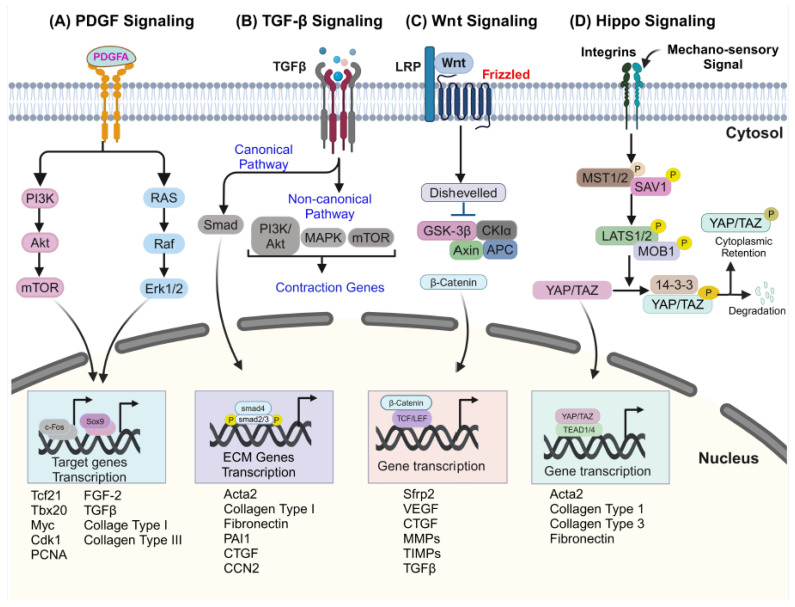
**Signaling pathways in cardiac fibroblast.** (**A**) Activated PDGFR stimulates the Ras pathway, which leads to the activation of RAF proto-oncogene serine/threonine protein kinase (Raf-1). Raf-1 activates downstream effectors, including dual-specificity mitogen-activated protein kinase (MEK) and extracellular signal-regulated protein kinase (ERK1/2) which promote the phosphorylation of several transcription factors to induce cell proliferation, cell survival, and ECM-related gene expression. (**B**) TGF-β signaling pathway. TGF-β homodimers phosphorylate and form a heterotetrameric complex with two type I receptors. The signaling domain of the type I receptor mediates phosphorylation and the activation of Smad proteins. The activated Smad complex forms a transcriptional module with several transcription factors, and co-factors to promote the transcription of ECM-related target genes. TGF-β activated the AKT pathway via a RhoA-dependent manner to induce the transcription of target genes. (**C**) Wnt signaling pathway. Wnt ligands bind to a frizzled receptor and co-receptor LRP. In the absence of a Wnt signal, cytoplasmic b-catenin phosphorylated by a multiprotein complex consisting of Axin, APC protein, and several kinases, is targeted for ubiquitin-mediated degradation. Upon Wnt binding, the destruction complex is dissembled, allowing β-catenin to translocate into the nucleus, where it binds with transcription factors (TCF/LEF) to regulate downstream gene expression. (**D**) Hippo signaling pathway. Various physiological or pathological signals induce the Hippo signaling pathway in fibroblasts. Upon activation, Hippo kinase complexes MST1/2, MOB1 SAV, and LATS1/2 induce phosphorylation of YAP and TAZ. Phosphorylated YAP and TAZ are sequestered in the cytoplasm by 14-3-3 proteins or targeted for degradation. The inactivation of the Hippo kinase complex results in the nuclear translocation of YAP and TAZ. In the nucleus, YAP and TAZ bind with TEAD factors to regulate gene expression.

**Figure 4 genes-16-00381-f004:**
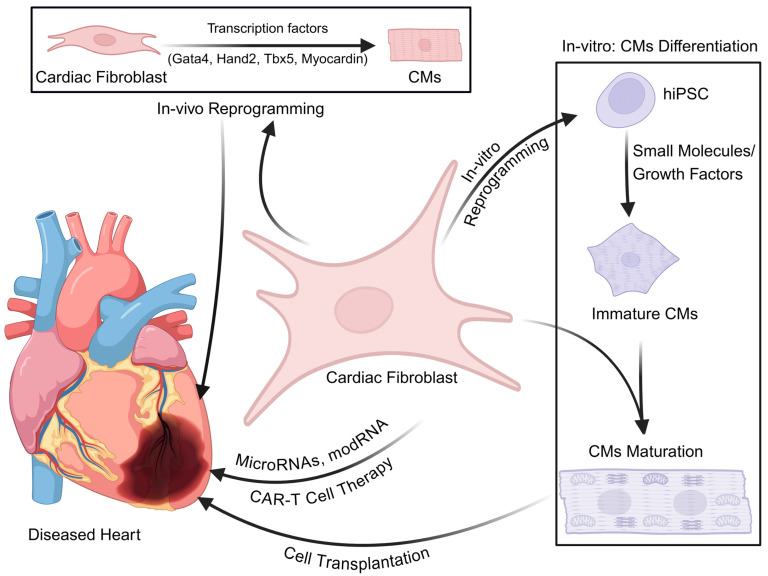
**Modifying cardiac fibroblasts for diverse applications.** Although detrimental to the injured heart tissue, recent studies have provided ways to utilize CFs for beneficial purposes. Direct reprogramming approaches have been tested using cardiogenic factors (Gata4, Hand2, Tbx5, myocardin) to cardiomyocytes both in vitro and in vivo. These methods have shown to reduced scarring with improved cardiac function. In the in vivo reprogramming, CFs differentiate into cardiomyocytes and mature to restore the damaged heart function [230]. Another strategy to utilize CF is to reprogram them to pluripotent stem cells using Yamanaka factors. These strategies have revolutionized cardiac regenerative medicine as well as developed cardiac disease models. In recent years, the reprogrammed CFs are being tested for personalized medicine. Cardiac fibroblast is also modified with *modifiedRNAs (modRNA)* or CAR-T cells to repair damaged hearts. Recently, CAR-T cell therapy has been used for cardiac fibrosis treatment [231]. Fibroblast activation protein (FAP), a glycoprotein expressed on a fibroblast surface during cardiac injury or fibrosis, are the target for the CAR-T cell approach to reduce cardiac fibrosis [232,233]. *MicroRNA (miRNA)* based therapy also modulates the cardiac fibroblast function to repair/regenerate the damaged heart [234,235]. In this approach, a combination of different *miRNAs* were used to regulate the proliferation and migration ability of cardiac fibroblast as well as activation of fibroblast into myofibroblast to repair the cardiac injury [236]. modRNA technology also targets specifically CFs to treat cardiac diseases [237]. Mice models of myocardial infarction showed that the cocktail of 7G modRNA treatment reduced cardiac scars as compared to the control and also reduced the collagen and fibronectin expression in treated mice [238].

**Table 1 genes-16-00381-t001:** Cardiac fibroblast markers at different developmental stages.

Markers/Factors	Embryonic Fibroblast	Neonatal Fibroblast	Adult Fibroblast	References
Tcf21	-	+	+	[22,25]
Tbx20	+	+	+	[26,27]
FSP1	-	+	+	[25,28]
Prolyl-4Hydroxylase	-	-	+	[29,30]
Vimentin	-	+	+	[31,32]
αSMA	-	-	-	[32,33]
PDGFRα	-	+	+	[2,31,34]
MEFSK4	-	-	+	[21]
DDR2	-	+	+	[25,35]
CD90	-	+	+	[21,36]
Sca1	-	-	+	[26,37]
Periostin	-	+	-	[25,32,38]
Fibronectin	-	+	+	[32,39]
Collagen type I	-	+	+	[25,31]
Collagen type III	-	+	+	[25]

**Table 2 genes-16-00381-t002:** Cardiac fibroblasts in cardiovascular diseases.

Disease	Gene Mutation	FB Activation	Outcomes	References
Hypertrophic Cardiomyopathy	*MYH7*, *MYBP3*	Activated	Contractile dysfunction due to the stiffening and thickening of the left ventricular, myocyte disarray and hypertrophy, heart failure or sudden cardiac death.	[122,123,124,125]
Dilated Cardiomyopathy	*TTN*, *LMN*, *EEF1A2*, *SYNE1*/*SYNE2*, *PRDM16*	Activated	Contractile dysfunction due to ventricular dilatation, heart failure or sudden cardiac death.	[123,126,127,128]
Duchene muscular dystrophy (DMD)	*Dystrophin*	Activated	Intracellular Ca^2+^ ion increases, myofiber atrophy/fibrosis, loss of muscle function, cardiac problems, respiratory dysfunctions, death between 20 and 40 years of age.	[129]
Becker muscular dystrophy (BMD)	*Dystrophin*	Activated	Milder than DMD and progresses slowly, Muscle weakness as in DMD.	[130]
Congenital Muscular Dystrophy (CMD)	*Merosin*, *LMNA*, *DAG1*	Activated	Joint contractures, cognitive and speech problems, seizures.	[131,132,133]

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
