# Peer review of "Cardiac Fibroblasts: Helping or Hurting"

_genes, 2025, doi:10.3390/genes16040381_

Round 1
Reviewer 1 Report
Comments and Suggestions for Authors
The title of review article is short and attractive since cardiac fibroblast/fibrocytes are permanent entity for investigation due to its implication in heart dysfunction (including heart failure) and arrhythmias (including stroke promoting atrial fibrillation and death promoting ventricular fibrillation). However, the text is less attractive in part due to long chapter Cardiac fibroblasts: Origin and Function without segmentation related to the topic.
Undouble, the issue dealing with the cardiac fibroblasts/fibrocytes and their research are in permanent progress to avoid fibrosis not only in the heart but also another organ like kidney, liver, etc to prevent life threatening stage. In this context question occurs what was the intention of authors, which category of readers would appreciate reviewed information’s about cardiac fibroblasts: students, clinicians, researchers?
I would suggest to include information preferably from recent papers with clear message to stimulate new ideas and further research. It is strange that authors mixed recent papers with old one e.g. 258 and 259, 261 and 262 or 264 and 265, etc. What was the reason for it? To see the progress in knowledge? Besides, it would be appreciated to include representative histological images of cardiac fibrosis demonstrating various types , e.g. interstitial, compact, diffuse, patchy, see please DeJong et al. 2011. This fact may complicate reviewed issue, what you think about?
Undoubtedly cardiac fibroblasts are permanently hot topic due to its vital-helping function that should be under control to prevent hurting.
Lastly, since less is more, question arises whether all detailed information and schematic figures should be included.
Author Response
Response to the reviewer#1
Thank you for the helpful comments regarding our manuscript (genes-3530293) entitled “Cardiac fibroblasts: Origin and Function”. As recommended by the reviewers, we have incorporated these changes in the revised review. Each issue raised by the panel is provided as a direct quote and is followed by our response.
Issue#1: The title of review article is short and attractive since cardiac fibroblast/fibrocytes are permanent entity for investigation due to its implication in heart dysfunction (including heart failure) and arrhythmias (including stroke promoting atrial fibrillation and death promoting ventricular fibrillation). However, the text is less attractive in part due to long chapter Cardiac fibroblasts: Origin and Function without segmentation related to the topic.
Response #1: As recommended by the reviewer, we have subdivided the sections into sub-sections in the revised manuscript. For example, section containing Cardiac fibroblasts: Origin and Function divided into two sections. Section 2.1 is focused on fibroblast origin, and Section 2.2 is focused on their functions. These sections covered the CFs origin and development as well as CFs crucial functions related to cardiac development and cardiac diseases. Similarly, we sub-divided the section 5: Strategies to mitigate Cardiac Fibrosis and Therapeutics into two sub-sections. Further, as suggested by the reviewer, we have shortened the text and rearranged the sentences throughout in the revised manuscript. Please see Section 2; page # 2-4 and Section 5; page # 15, 16 in the revised manuscript.
Issue#2: Undouble, the issue dealing with the cardiac fibroblasts/fibrocytes and their research are in permanent progress to avoid fibrosis not only in the heart but also another organ like kidney, liver, etc to prevent life threatening stage. In this context question occurs what was the intention of authors, which category of readers would appreciate reviewed information’s about cardiac fibroblasts: students, clinicians, researchers?
Response #2: In this review article, we have comprehensively reviewed cardiac fibroblast biology in terms of their origin, functions, signaling mechanism involved in their regulation, their role in disease manifestation, and their therapeutic applications. Further, we have added future perspective to provide further understanding of the cardiac fibroblast biology during cardiac development and disease in the revised manuscript. Please see page # 2 (lines; 73-75), #5 (lines; 160-163), #8 (lines; 231-233), #9 (lines; 293-294), #12 (lines; 394-396), #13 (lines; 471-473), #15 (lines; 530-532). We believe that this review will be helpful for a broader cardiovascular scientific community, including students, clinicians as well as researchers.
Issue#3: I would suggest to include information preferably from recent papers with clear message to stimulate new ideas and further research. It is strange that authors mixed recent papers with old one e.g. 258 and 259, 261 and 262 or 264 and 265, etc. What was the reason for it? To see the progress in knowledge? Besides, it would be appreciated to include representative histological images of cardiac fibrosis demonstrating various types , e.g. interstitial, compact, diffuse, patchy, see please DeJong et al. 2011. This fact may complicate reviewed issue, what you think about?
Response #3: As suggested by the reviewer, we have added several new sentences and future perspective to provide further understanding of the cardiac fibroblast biology during cardiac development and disease in the revised manuscript. Please see page # 2 (lines; 73-75), #5 (lines; 160-163), #8 (lines; 231-233), #9 (lines; 293-294), #12 (lines; 394-396), #13 (lines; 471-473), #15 (lines; 530-532). This also encompasses the prospective new ideas based on our review of the literature. We agree with the reviewer with the types and extent of cardiac fibrosis. As suggested by the reviewer, we have reviwed the article from DeJong et al. 2011 and added the types of fibrosis in the revised manuscript. Please see page#5; lines 146-149. Further, we agree with the reviewer that adding detailed information about cardiac fibrosis will not only lengthen the review article but will also complicate it. Therefore, we have not added new figure in the revised manuscript.
Issue#4: Undoubtedly cardiac fibroblasts are permanently hot topic due to its vital-helping function that should be under control to prevent hurting.
Response #4: We agree with the reviewer and thank you for the positive remarks.
Issue#5: Lastly, since less is more, the question arises whether all detailed information and schematic figures should be included.
Response #5: As suggested by the reviewer, we have shortened the sections throughout the revised article, however, we think that the figures provide more clarity and mechanistic understanding. As recommended by reviewer#2, we have further provided more detailed functions of cardiac fibroblast, which are now incorporated in the Figure 1 and corresponding Figure legend for more clarity and alignment with the text.
Reviewer 2 Report
Comments and Suggestions for Authors
The review article by Shameem et al., titled "Cardiac Fibroblasts: Helping or Hurting," offers a detailed examination of cardiac fibroblasts (CFs) and their roles in normal heart development and various cardiac diseases. It is written exceptionally well. It discusses the embryological origins of CFs, the intricate signaling pathways that govern their function, and their dual role in tissue repair and pathological fibrosis. Additionally, the manuscript reviews current therapeutic approaches aimed at modifying fibroblast activity for cardiac repair. Overall, the subject is timely and significant; however, there are several areas that could benefit from enhancements to improve clarity, consistency, and overall impact.
Major Points for Improvement:
- Although the review is comprehensive, some transitions between sections (e.g., from developmental biology to signaling pathways and therapeutic strategies) are abrupt. Consider adding bridging paragraphs or summary tables to connect the topics clearly and guide the reader.
- The section on treatment options could benefit from a more structured approach. For instance, separating pharmacological interventions from gene- and cell-based therapies (including reprogramming and immunotherapeutic approaches) would help readers appreciate the range of strategies and their respective challenges.
- While the article summarizes extensive literature, a more critical discussion is needed. Consider highlighting controversies (e.g., the dual “friend or foe” role of CFs) along with the limitations or gaps in current research. A dedicated section on unresolved questions and future directions would enhance the article's overall impact review.
- The manuscript mentions several figures (Figures 1–4) and tables (e.g., Table 1 and Table 2). Ensure that all figures include clear legends, high-resolution images, and consistent formatting.
- Consider adding summary diagrams or flowcharts to integrate the multiple signaling pathways discussed. A comprehensive table summarizing the key markers, origins, and roles of CFs in different cardiac conditions could also be very useful.
Minor Points and Technical Issues:
-
- There are several instances where words are broken by hyphenation due to line breaks (e.g., “ex- periments,” “Iin vitro”). These should be corrected so that words appear intact.
- In line 38, the author refers to [4]. However, Abercrombie and colleagues published five separate manuscripts on fibroblast biology. It is suggested that the authors use appropriate references to back up their claims.
- Figure 1 requires improvement to connect with the text, such as incorporating the EMT and EDFP processes.
- Lines 71-74 must include the relevant reference.
- In lines 83-84, this line needs improvement regarding the interaction between pericytes and fibroblasts.
Author Response
Response to Reviewer#2
Thank you for finding this review article as a valuable source as it offers a detailed examination of cardiac fibroblasts (CFs) and their roles in normal heart development and various cardiac diseases. Each issue raised by the panel is provided as a direct quote and is followed by our response.
Major Comments
Issue#1: Although the review is comprehensive, some transitions between sections (e.g., from developmental biology to signaling pathways and therapeutic strategies) are abrupt. Consider adding bridging paragraphs or summary tables to connect the topics clearly and guide the reader.
Response #1: As recommended by the reviewer, for more clarity, we have divided the sections into sub-sections in the revised manuscript. For example, section containing Cardiac fibroblasts: Origin and Function, is divided into two sections. Section 2.1 is focused on fibroblast origin, and Section 2.2 is focused on their functions. Similarly, we divided the section 5: Strategies to mitigate Cardiac Fibrosis and Therapeutics into two sub-sections. As recommended by the reviewer#1 and #2, we have added several new sentences and future perspective to provide further understanding of the cardiac fibroblast biology during cardiac development and disease in the revised manuscript. Please see page # 2 (lines; 73-75), #5 (lines; 160-163), #8 (lines; 231-233), #9 (lines; 293-294), #12 (lines; 394-396), #13 (lines; 471-473), #15 (lines; 530-532). This also encompasses the prospective new ideas based on our review of the literature. Further, we have shortened the text and rearranged the sentences throughout in the revised manuscript. Please see Section 2; page numbers 2-6 and Section 5; page numbers 15, 16 in the revised manuscript.
Issue#2: The section on treatment options could benefit from a more structured approach. For instance, separating pharmacological interventions from gene- and cell-based therapies (including reprogramming and immunotherapeutic approaches) would help readers appreciate the range of strategies and their respective challenges.
Response #2: As suggested by the reviewer, we have subdivided this section into two sections. we divided the section 5: Strategies to mitigate Cardiac Fibrosis and Therapeutics into two sub-sections for a more structure approach. Please see page# 15, 16 in the revised manuscript.
Issue#3: While the article summarizes extensive literature, a more critical discussion is needed. Consider highlighting controversies (e.g., the dual “friend or foe” role of CFs) along with the limitations or gaps in current research. A dedicated section on unresolved questions and future directions would enhance the article's overall impact review.
Response #3: As recommended by the reviewer#1 and #2, we have added several new sentences and future perspective to provide further understanding of the cardiac fibroblast biology during cardiac development and disease in the revised manuscript. Please see page # 2 (lines; 73-75), #5 (lines; 160-163), #8 (lines; 231-233), #9 (lines; 293-294), #12 (lines; 394-396), #13 (lines; 471-473), #15 (lines; 530-532). This also encompasses the prospective new ideas based on our review of the literature. Further, we have shortened the text and rearranged the sentences throughout in the revised manuscript. Please see Section 2; page numbers 2-6 and Section 5; page numbers 15, 16 in the revised manuscript.
Issue#4: The manuscript mentions several figures (Figures 1–4) and tables (e.g., Table 1 and Table 2). Ensure that all figures include clear legends, high-resolution images, and consistent formatting.
Response #4: As suggested by the reviewer, we have carefully crafted the figures and provided high-resolution images with consistent formatting.
Issue#5: Consider adding summary diagrams or flowcharts to integrate the multiple signaling pathways discussed. A comprehensive table summarizing the key markers, origins, and roles of CFs in different cardiac conditions could also be very useful.
Lastly, since less is more, question arises whether all detailed information and schematic figures should be included.
Response #5: As noted in the review article, we have several figures (Figures 1-4) and two tables (Table 1 and Table 2). In particular, Figure 3 integrates the multiple signaling pathways and downstream effectors. Also, the tables provide comprehensive information about cardiac fibroblast biology during development and disease conditions. Further, as suggested by the reviewer#1 and #2, “less is more,” we have not included any new figures in the revised manuscript.
Minor Comments
Issue#1: There are several instances where words are broken by hyphenation due to line breaks (e.g., “ex- periments,” “Iin vitro”).
Response #1: Thank you for pointing out the errors. As suggested by the reviewer, we have corrected the words in the revised manuscript.
Issue#2: In line 38, the author refers to [4]. However, Abercrombie and colleagues published five separate manuscripts on fibroblast biology. It is suggested that the authors use appropriate references to back up their claims.
Response #2: As suggested by the reviewer, we have incorporated the appropriate reference to support the text. Please see page #1 in the revised manuscript.
Issue#3: Figure 1 requires improvement to connect with the text, such as incorporating the EMT and EDFP processes.
Response #3: As suggested by the reviewer, we have incorporated this information in the revised manuscript. Please see new Figure 1 in the revised manuscript.
Issue#4: Lines 71-74 must include the relevant reference.
Response #4: As suggested by the reviewer, we have incorporated the appropriate references for the given sentences. Please see page 2, line 67 in the revised manuscript.
Issue#5: In lines 83-84, this line needs improvement regarding the interaction between pericytes and fibroblasts.
Response #5: We agree with the reviewer’s comment. Since this section focuses on cardiac fibroblast origin, we refrained from providing information related to cellular interactions. To avoid confusion, we have removed these sentences in the revised manuscript.